# IN YOUR PACE: LEARNING THE RIGHT EXAMPLE AT THE RIGHT TIME

## ABSTRACT

Training neural networks is traditionally done by sequentially providing random mini-batches sampled uniformly from the entire dataset. In our work, we show that sampling mini-batches non-uniformly can both enhance the speed of learning and improve the final accuracy of the trained network. Specifically, we decompose the problem using the principles of curriculum learning: first, we sort the data by some difficulty measure; second, we sample mini-batches with a gradually increasing level of difficulty. We focus on CNNs trained on image recognition. Initially, we define the difficulty of a training image using transfer learning from some competitive "teacher" network trained on the Imagenet database, showing improvement in learning speed and final performance for both small and competitive networks, using the CIFAR-10 and the CIFAR-100 datasets. We then suggest a bootstrap alternative to evaluate the difficulty of points using the same network without relying on a "teacher" network, thus increasing the applicability of our suggested method. We compare this approach to a related version of Self-Paced Learning, showing that our method benefits learning while SPL impairs it.

## 1 INTRODUCTION

Teaching complex tasks to humans and animals can be difficult. Often such tasks cannot be grasped by the learners, or "students", immediately, and need to be broken down into simpler problems. Therefore, in order to teach complex tasks, teachers are often required to create a curriculum. The curriculum imposes some order on the learning task; it introduces different concepts at different times, hence exploiting previously learned concepts in order to ease the abstraction of new ones. Imposing a curriculum in order to speed up learning is widely used in the context of human learning, and also routinely used in animal training (Skinner, 1958; Pavlov, 2010; Krueger & Dayan, 2009).

In many traditional machine learning approaches, known as supervised learning, a target function is estimated by using a set of labeled examples. The examples can be thought of as given by a teacher while the learning algorithm can be thought of as a student. The field of curriculum learning (CL), which is motivated by the idea of a curriculum in human learning, attempts at imposing some structure on the labeled data. Such structure essentially relies on a notion of "easy" and "hard" examples and utilizes this distinction in order to teach the student how to generalize easier concepts before harder ones. Empirically, the use of CL has been shown to accelerate and improve the learning process (e.g. Selfridge et al., 1985; Bengio et al., 2009).

In this work, we aim at extending the understanding of CL in the context of deep neural learning. More specifically, we wish to understand to what extent curriculum can improve the accuracy and convergence rate of deep neural networks. The main challenge in making CL practical is, arguably, finding a way to construct a good curriculum for a newly unseen database. In order to do so, we investigate two ideas motivated by transfer learning and bootstrapping respectively.

When establishing a curriculum for human students, teachers need to arrange the material in a way that will present simple concepts before harder ones, so that the abstraction of simple ideas can help the student grasp more complex ones (Hunkins & Ornstein, 2016). However, sorting the concepts by difficulty is not sufficient. The teacher also needs to attend to the pace by which the material is presented – going over the simple ideas too fast may lead to more confusion than benefit, while moving along too slowly may lead to boredom and unproductive learning (Hunkins & Ornstein, 2016). These principles can also be beneficial when the learner is a neural network.

Specifically, formalizing and generalizing what was implicitly done in Weinshall et al. (2018), we decompose the problem of CL and define two separate - but closely related - functions. The first function, termed *scoring function*, determines the "hardness" or "complexity" of each example in the data. The *scoring function* enables us to sort the data by concept difficulty, allowing us to present to the network the easier (and presumably simpler) examples first. The underlying assumption is that generalization from the easier examples can simplify the learning of harder examples in the data. The second function, termed *pacing function*, determines the pace by which data is presented to the network. The pace depends on both the data itself and the learner.

In our work, we analyze several *scoring* and *pacing* functions, investigating their inter-dependency and presenting ways to combine them in order to achieve faster learning and better generalization. The main challenge is, arguably, how to obtain an effective *scoring function* without additional human supervision. To this end we investigate two approaches, each providing a different estimator for the ideal *scoring function*: (i) Knowledge transfer. The first *scoring function* is based on transfer learning from networks trained on the large and versatile Imagenet dataset (Deng et al., 2009; Weinshall et al., 2018). (ii) Bootstrapping. The second *scoring function* is based on self-tutoring - we train the network once without curriculum, then use the resulting classifier to rank the training data in order to train the same network again from scratch. Both *scoring functions* are shown in Section 3 to speed up learning and improve the generalization of neural networks.

In many approaches, including Self-Paced Learning (SPL), Active-Learning and hard example mining (Kumar et al., 2010; Schein & Ungar, 2007; Shrivastava et al., 2016), the mini-batches which presented to the learner model are as sampled dynamically, based at each time point on the current hypothesis of the model. While in some contexts these approaches are beneficial (Chang et al., 2017; Zhang et al., 2017), they are based on the knowledge of the student at a specific time point. While a student can report what is easy/hard for it right now, it might be oblivious to some aspects of the bigger problem at hand, ignoring concepts which if learned early, could prove helpful in a later time. In the context of linear regression loss, Weinshall et al. (2018) showed that such distinction indeed holds: while it is beneficial to prefer points with **lower loss** with respect to the target hypothesis as suggested by CL, it is on the other hand beneficial to prefer points with **higher loss** with respect to the current hypothesis in agreement with hard data mining (Shrivastava et al., 2016) and boosting, contrary to SPL.

To examine this somewhat confusing point, we have implemented a simplified version of the procedure described above, where the *scoring function* is based on the loss of the training points with respect to the current hypothesis, both in ascending and descending orders. These variants of SPL and hard example mining respectively learn slower and reach lower final accuracy when compared to self-taught, throughout all of our experiments.

We have also investigated three *pacing functions*. (i) *Fixed exponential pacing* presents the learner initially with a small percentage of the data, increasing the amount exponentially every fixed number of learning iterations. (ii) *Varied exponential pacing* allows the number of iterations in each step to vary as well. (iii) *Single-step pacing* is a simplified version of the first protocol, where mini-batches are initially sampled from a fixed fraction of the data that includes the easiest examples, after which mini-batches are sampled from the whole data as usual. We show that the three functions have comparable performance, and analyze the complexity of their use.

**Previous work.** While remaining in the fringes of machine learning, there has been some recent work on CL and its applications. Bengio et al. (2009) introduced the idea of CL for machine learning algorithms, showing simple examples where CL benefits learning. Weinshall et al. (2018) proved that CL boosts the speed of convergence in the convex case of linear regression. Otherwise most prior art is empirical, and almost always ranking by difficulty (i.e., the *scoring function* defined above) is provided by the user based on prior knowledge (in other words, supervision) as in Jesson et al. (2017).

In a closely related line of works, a pair of teacher and student networks are trained simultaneously, where mini-batches for the student network are sampled dynamically by the teacher, based on the student's output in each time point (Jiang et al., 2018; Fan et al., 2018). As opposed to our method, these works base the curriculum on the current hypothesis of the students, and achieve better performance for corrupted (Jiang et al., 2018) or smaller (Fan et al., 2018) datasets, instead of improved generalization on the original dataset.

**Our contribution**, with respect to this previous work, is to provide a formal definition of CL algorithms by way of 2 functions for scoring and pacing, analyze and comparatively evaluate these functions, and show how CL can benefit learning in CNNs even without human supervision about the ranking of examples by difficulty and in a problem-free manner.

## 2 CURRICULUM LEARNING

Curriculum learning deals with the question of **how to use prior knowledge about the difficulty of the training examples**, in order to sample each mini-batch non-uniformly and thus boost the rate of learning and the accuracy of the final classifier. The paradigm of CL is based on the intuition that it helps the learning process when the learner is presented with simple concepts first.

### 2.1 NOTATIONS AND DEFINITIONS

Let $\mathbb{X} = \{(x_i, y_i)\}_{i=1}^N$ denote the data, where $x_i \in \mathbb{R}^d$ denotes a single data point and $y_i \in [K]$ its corresponding label. Let $F_\theta : \mathbb{R}^d \to [K]$ denote the target classifier (or learner), and mini-batch $\mathbb{B} \subseteq \mathbb{X}$ denote a subset of $\mathbb{X}$. In the most common training procedure, which is a robust variant of Stochastic Gradient Descent (SGD), $F_\theta$ is trained sequentially when given as input a sequence of mini-batches $[\mathbb{B}_1, ..., \mathbb{B}_M]$ (Shalev-Shwartz & Ben-David, 2014). The common approach – denoted *vanilla* in the following sections – samples each mini-batch $\mathbb{B}_i$ uniformly from $\mathbb{X}$. Both in the common approach and in our work, the size of each mini-batch remains constant, to be considered as a hyper-parameter defining the learner.

We measure the difficulty of point $x_i$ by its minimal loss with respect to the set of optimal hypotheses under consideration. We define a *scoring function* (or a "*difficulty*" function) to be any function $f : \mathbb{X} \to \mathbb{R}$, and say that example $(x_i, y_i)$ is more "difficult" than example $(x_j, y_j)$ if $f(x_i, y_i) > f(x_j, y_j)$. Choosing $f$ is the main challenge of CL, as it encodes the prior knowledge of the teacher.

We define a *pacing function* to be a function $g_{F_\theta} : [M] \to [N]$, which may depend on the learner $F_\theta$. The *pacing function* is used to determine a sequence of subsets $\mathbb{X}_1', ..., \mathbb{X}_M' \subseteq \mathbb{X}$, of size $|\mathbb{X}_i'| = g_{F_\theta}(i)$, from which $\{\mathbb{B}_i\}_{i=1}^M$ are sampled uniformly. In CL the $i$-th subset $\mathbb{X}_i'$ includes the first $g_{F_\theta}(i)$ elements of the training data when sorted by the *scoring function* $f$ in an ascending order. Although the choice of the subset can be encoded in the distribution from which each $\mathbb{B}_i$ is sampled, adding a *pacing function* simplifies the exposition and analysis.

### 2.2 CURRICULUM LEARNING METHOD

Together, each *scoring function* $f$ and *pacing function* $g_{F_\theta}$ define a curriculum. Any learning algorithm which uses the ensuing sequence $[\mathbb{B}_i]_{i=1}^M$ is a **curriculum learning algorithm**. We note that in order to avoid bias when picking a subset of the $N$ examples for some $N$, it is important to keep the sample balanced with the same number of examples from each class as in the training set. Pseudo-code for the CL algorithm is given in Alg. 1.

In order to narrow down the specific effects of using a *scoring function* based on ascending difficulty level, we examine two control conditions. Specifically, we define 2 additional *scoring functions* and corresponding algorithms: (i) The **anti-curriculum algorithm** uses the *scoring function* $f' = -f$, where the training examples are sorted in a *descending order*; that results in presenting the harder examples before the easier ones. (ii) The **random-curriculum algorithm** (henceforth denoted random) uses a *scoring function* where the training examples are randomly sorted.

### 2.3 SCORING AND PACING FUNCTIONS

We evaluate two *scoring functions*: (i) *Transfer scoring function*, computed as follows: First, take the pre-trained Inception network (Szegedy et al., 2016) and run each training image through it, using the activation levels of its penultimate layer as a feature vector (Caruana, 1995). Second, use these

features to train a classifier and use its confidence score as the *scoring function* for each image[1]. (ii) *Self-taught scoring function*, computed as follows: First, train the network using uniformly sampled mini-batches (the *vanilla* method). Second, compute this network's confidence score for each image to define a *scoring function*[2].

Although the *pacing function* can be any function $g_{F_\theta} : [M] \to [N]$, we limit ourselves to monotonic increasing functions so that the likelihood of the easier examples can only decrease. For simplicity, $g_{F_\theta}$ is limited to staircase functions. Thus each *pacing function* is defined by the following hyper-parameters, where *step* denotes all the learning iterations during which $g_{F_\theta}$ remains constant: *step_length* - the number of iterations in each *step*; *increase* - an exponential factor used to increase the size of the data used for sampling mini-batches in each *step*; *starting_percent* - the fraction of the data in the initial *step*. An illustration of these parameters can be seen in Fig. 1.

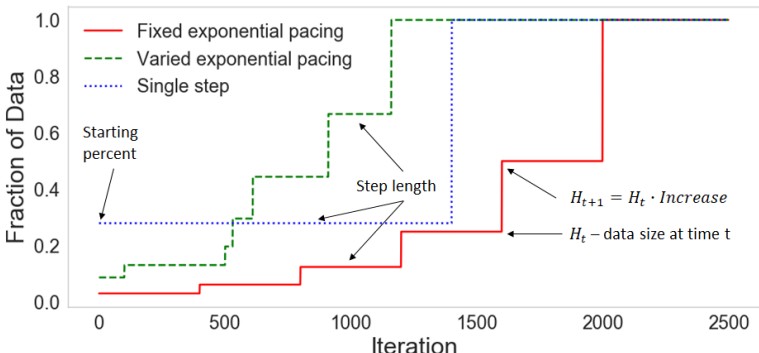

Figure 1: Illustration of the 3 *pacing functions* used below, showing the different hyper-parameters that define each of them (see text). The values of the hyper-parameters used in this illustration were chosen arbitrarily, for illustration only.

We evaluate three *pacing functions*: (i) *Fixed exponential pacing* has a fixed *step_length*, and exponentially increasing data size in each *step*. Formally, the *pacing function* is given by:

$$g_{F_\theta}(i) = \min\left(starting\_percent \cdot increase^{\lfloor \frac{i}{step\_length} \rfloor}, 1\right) \cdot N$$

---

[1]Similar results can be obtained when using different confidence scores (e.g, the classifier's margin), different classifiers (e.g, linear SVM), and different teacher networks (e.g, VGG-16 (Simonyan & Zisserman, 2014), Resnet (He et al., 2016)). For more details, see Appendix A.

[2]Theoretically we can use this method repeatedly, as discussed in Appendix B.

---

**Algorithm 1:** Curriculum learning method

    **Input** : *pacing function* $g_{F_\theta}$, *scoring function* $f$, labeled data $\mathbb{X}$.

    **Output:** sequence of mini-batches $\left[\mathbb{B}'_1, ..., \mathbb{B}'_M\right]$.

**1** sort $\mathbb{X}$ according to $f$, in ascending order;

**2** $result \leftarrow []$;

**3 for** $i = 1, ..., M$ **do**

**4**      $size \leftarrow g_{F_\theta}(i)$;

**5**      $\mathbb{X}'_i \leftarrow \mathbb{X}[1, ..., size]$;

**6**      uniformly sample $\mathbb{B}'_i$ from $\mathbb{X}'$;

**7**      append $\mathbb{B}'_i$ to $result$;

**8 end**

**9 return** result;

(ii) *Varied exponential pacing*, which allows *step_length* to vary as well[3]:

$$g_{F_\theta}(i) = \min\left(starting\_percent \cdot increase^{\sum_{k=1}^{\#steps} \mathbf{1}[i>step\_length_k]}, 1\right) \cdot N$$

The total number of *steps* can be calculated from *starting_percent* and *increase*:

$$\#step = \lceil -\log_{increase}(starting\_percent)\rceil$$

(iii) *Single step pacing*, which is a simplification of the staircase function into a step function:

$$g_{F_\theta}(i) = starting\_percent^{\mathbf{1}[i<step\_length]} \cdot N$$

This function has only 2 hyper-parameters, hence it is simpler to use than the previous two.

## 3 EMPIRICAL EVALUATION

**Methodology.** All the code used in this work will be published upon acceptance. We define 4 empirical cases: **Case 1** replicates the experimental design described in (Weinshall et al., 2018), by using the same dataset and network architecture. The dataset is the "small mammals" superclass of CIFAR-100 (Krizhevsky & Hinton, 2009), containing a subset of 3000 images from CIFAR-100, divided into 5 classes of small mammals (hamster, mouse, rabbit, shrew, squirrel). Each class contains 500 training images and 100 test images. The neural network is a moderate size hand-crafted convolutional network, whose architecture details can be found in Appendix C. **Cases 2** and **3** adopt the same architecture used above while being applied to the entire CIFAR-10 and CIFAR-100 datasets, where the network's output layer is adjusted to size 10 and 100 respectively. **Case 4** uses a public-domain VGG-based architecture[4], which achieves competitive results (Simonyan & Zisserman, 2014; Liu & Deng, 2015), to classify the CIFAR-100 dataset.

**Hyper-parameter tuning.** As in all empirical studies involving deep learning, the results are quite sensitive to the values of the hyper-parameters, hence parameter tuning is required. Issues related to how a fair comparison between the different conditions is achieved are discussed in Appendix B. In practice, in order to reduce the computation time of parameter tuning, we varied only the first 2 *step_length* instances in the *varied exponential pacing* condition. Accordingly, *fixed exponential pacing*, *varied exponential pacing* and *single step pacing* define 3, 5 and 2 new hyper-parameters respectively, referred to henceforth as the *pacing* hyper-parameters.

In the CL framework, the use of a *pacing function* affects the optimal values of other hyper-parameters, in particular, the learning rate. Specifically, since it significantly reduces the size of the data-set from which each mini-batch is sampled, this has the concomitant effect of increasing the effective learning rate. As a result, when using the *fixed exponential* or the *single step pacing functions*, the learning rate must be tuned separately for every test condition. As traditionally done (e.g Simonyan & Zisserman, 2014; Szegedy et al., 2016; He et al., 2016), we set an initial learning rate and decrease it exponentially every fixed number of iterations. This method gives rise to 3 learning rate hyper-parameters which require tuning: (i) the initial learning rate; (ii) the factor by which the learning rate is decreased; (iii) the length of each step with constant learning rate[5].

When *varied exponential pacing* is used, varying *step_length* has the opposite concomitant effect on the learning rate, as it determines the number of mini-batch samples in each *step*. Effective tuning of this parameter can make the additional tuning of parameters affecting the learning rate redundant. In practice, in order to reach the improvement achieved by the *fixed exponential pacing*, we decrease the corresponding learning rate parameters used in the *vanilla* condition by some small factor[6].

### 3.1 RESULTS: CL BENEFITS LEARNING

**Case 1:** A moderate size network is trained to distinguish 5 classes from CIFAR-100, which are members of the same super-class as defined in the original dataset. Results are shown in Fig. 2.

---

[3]In practice, to avoid an unfeasible need to tune too many hyper-parameters, we vary only the first two *step_length* instances and fix the rest. As shown later on, this is reasonable as most of the power of the curriculum lies in the first few *steps*.

[4]The code for the VGG network is available at https://github.com/geifman/cifar-vgg.

[5]For more details, see Appendix B.

[6]In the results reported below we used a reduction of 10%, with similar behavior for other nearby choices.

Curriculum learning is clearly and significantly beneficial - learning starts faster, and converges to a better solution. We observe that the performance of CL with a random *scoring function* is similar to *vanilla*, indicating that the main reason for the improvement achieved by CL is due to its beneficial *transfer scoring function*. In fact, although tuned separately, the learning rate hyper-parameters for both the random and the curriculum test conditions are very similar, confirming that the improved performance is due to the use of an effective *transfer scoring function*.

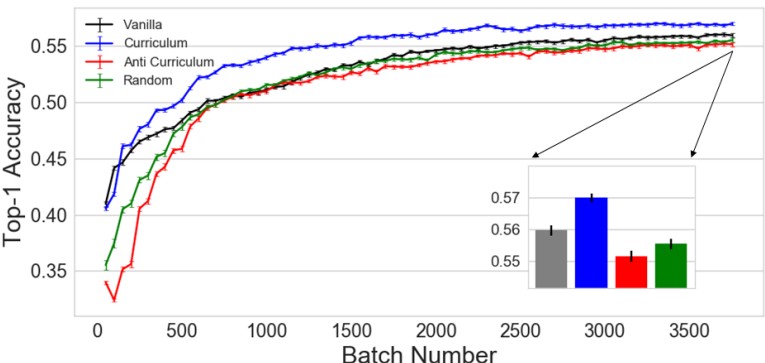

Figure 2: Results in **case 1**, with Inception-based *transfer scoring function* and *fixed exponential pacing function*. Inset: bars indicating the average final accuracy in each condition over the last few iterations. Error bars indicate the STE (STandard Error of the mean) after 50 repetitions. The curriculum method (in blue) reaches higher accuracy faster, and converges to a better final solution.

To check the robustness of these results, we repeated the same empirical evaluation using different super-classes of CIFAR-100, with similar results (see Appendix A). Interestingly, we note that the observed advantage of CL is more significant when the task is more difficult (i.e. lower *vanilla* test accuracy). The reason may be that in easier problems there is a sufficient number of easy examples in each mini-batch even without CL. Although the results reported here are based on transfer from the Inception network, we are able to obtain the same results using *scoring functions* based on transfer learning from other large networks, including VGG-16 and Resnet, as shown in Appendix A.

**Cases 2 and 3:** Similar empirical evaluation as in **case 1**, using the same moderate size network to classify two benchmark datasets. The results are shown in Fig. 3. Like before, the test accuracy in the curriculum test condition increases faster and achieves better final performance in both cases, as compared to the *vanilla* test condition. The beneficial effect of CL is larger when classifying the CIFAR-100 dataset, which is a harder dataset.

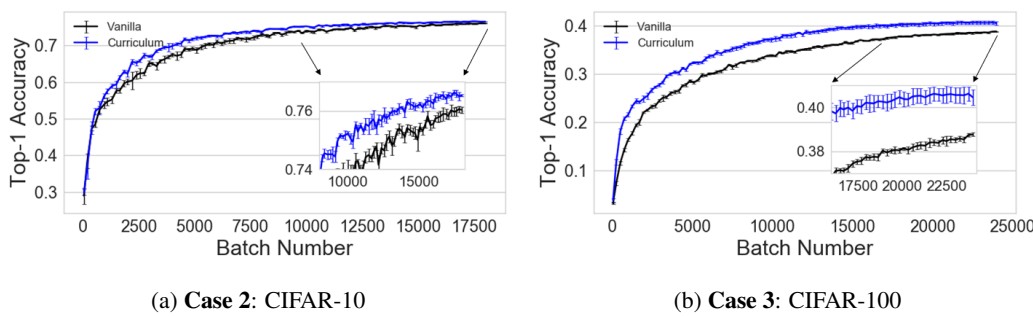

(a) **Case 2**: CIFAR-10                                      (b) **Case 3**: CIFAR-100

Figure 3: Results in **cases 2** and **3**, with Inception-based *transfer scoring function* and *fixed exponential pacing function*. Inset: zoom-in on the final iterations, for better visualization. Error bars show STE after 5 repetitions. (a) CIFAR-10 dataset, (b) CIFAR-100 dataset.

**Case 4:** Similar empirical evaluation as in **case 1**, using a competitive public-domain architecture. Specifically, we use the Inception-based *transfer scoring function* to train a VGG-based network

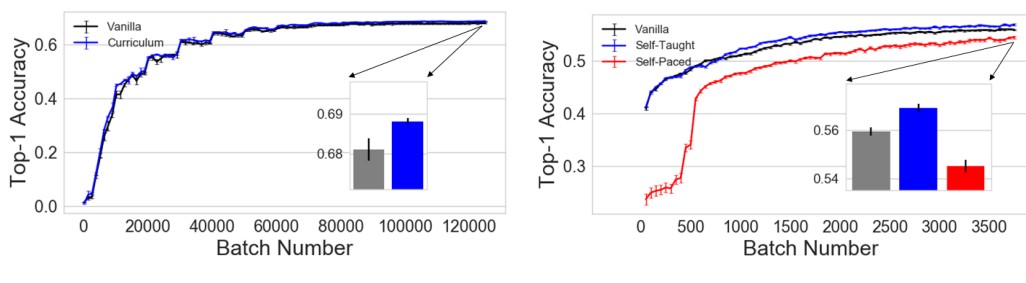

(a) CIFAR100, VGG-based network  (b) *Self-taught learning* vs. self-paced learning.

Figure 4: (a) Curriculum learning using a competitive VGG-based network running on the entire CIFAR-100 dataset. (b) Results in **case 1** with the Inception-based *transfer scoring function*, showing both the *self-taught scoring function* and the *self-paced scoring function*. Inset: bars indicating the average final accuracy in each condition over the last few iterations. Error bars indicate the STE after 3 repetitions in (a) and 50 repetitions in (b).

(Liu & Deng, 2015) to classify the CIFAR-100 dataset. Differently from the previous cases, here we use the *varied exponential pacing function* with a slightly reduced learning rate, as it has the fewest hyper-parameters to tune, an important factor when training such a big network. Results are shown in Fig. 4a (with no data augmentation), showing the same qualitative results as in the previous cases; CL gives a smaller benefit, but the benefit is still significant.

**Case 5:** Similar empirical evaluation as in **case 1**, using the same moderate size network to distinguish 7 classes of cats from the ImageNet dataset[7]. The results are shown in Fig. 5. Again, the test accuracy in the curriculum test condition increases faster and achieves better final performance in the curriculum case, as compared to the *vanilla* test condition.

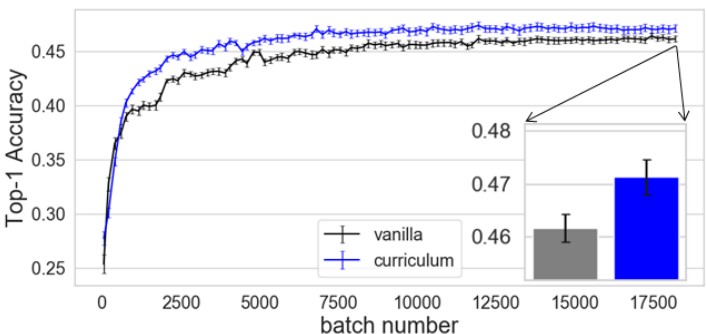

Figure 5: Results in **case 5**, with Inception-based *transfer scoring function* and *fixed exponential pacing function*. Inset: zoom-in on the final iterations, for better visualization. Error bars show STE after 25 repetitions.

## 3.2 SELF-TAUGHT CURRICULUM LEARNING VS. SELF-PACED LEARNING

Curriculum learning is closely related to the idea of Self-Paced Learning (SPL), an iterative procedure where higher weights are given to training examples that have lower cost with respect to the current hypothesis. In fact, SPL may appear similar, or closely related, to the idea of *self-taught* learning. The main difference between the methods is that self-paced learning determines the *scoring function* according to the loss with respect to the current hypothesis (or network), while the *self-taught scoring function* is based on the loss with respect to the final hypothesis of a trained network. In accordance, we define the *self-paced scoring function*, where each point is scored by its

---

[7]For more details, see Appendix C

loss with respect to the current network. Note that when using CL to optimize the linear regression loss (see introduction), *self-taught curriculum* and self-paced learning are discordant.

To compare the *self-taught scoring function* and the *self-paced scoring function*, we investigate their effect on CL in the context of empirical **case 1**. Results are shown in Fig. 4b. As expected, we see that CL using the *self-taught scoring function* improves the test accuracy throughout the entire learning session. On the other hand, CL training using the *self-paced scoring function* decreases the test accuracy throughout. This decrease is more prominent at the beginning of the learning, where most of the beneficial effects of the curriculum are observed, suggesting that the *self-paced scoring function* can significantly delay learning.

### 3.3 THE SCORING FUNCTION: ANALYSIS AND EMPIRICAL EVALUATION

In order to analyze the effects of transfer based *scoring functions*, we turn to analyze the gradients of the network's weights w.r.t the empirical loss. We evaluate the gradients using a pre-trained *vanilla* network in the context of **case 1**. First, for each method and each *scoring function*, we collect the subset of points used to sample the first mini-batch according to the *pacing function* $g_{F_\theta}(1)$[8]. For comparison, we also consider the set of all training points, which are used to compute the exact gradient of the empirical loss in batch learning using GD. We then compute the corresponding set of gradients for the training points in each of these subsets of training points, treating each layer's parameters as a single vector, and subsequently estimate the gradients' mean and total variance[9], used to evaluate the coherence of the gradients in the first mini-batch of each *scoring function*. The Euclidean distance between the mean gradient in the different conditions is used to estimate the similarity between the different *scoring functions*, based on the average preferred gradient.

We can now compare the set of gradients thus defined using three *transfer scoring functions*, which differ in the parent network used for scoring the points: 'VGG-16', 'Resnet', and 'Inception'. We include in the comparison the gradients of the *random scoring function* denoted 'Random', and the gradients of the whole batch of training data denoted 'All'. Results are shown in Fig. 6.

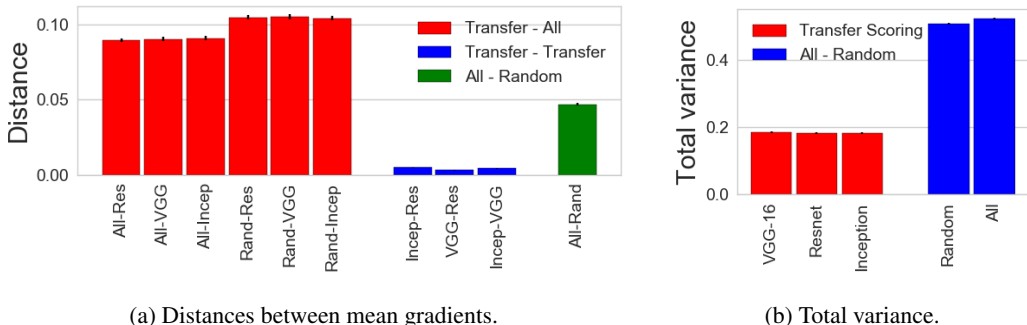

(a) Distances between mean gradients.      (b) Total variance.

Figure 6: (a) Distance between the mean gradient direction of preferred examples under different *scoring functions*. Each bar corresponds to a pair of mean gradients in two different conditions, see details in the main text. (b) The total variance of each set of gradients.

We see in Fig. 6a - blue bars - that the average gradient vectors, computed based on the 3 *transfer scoring functions*, are quite similar to each other. This suggests that they are pointing towards nearby local minima in parameters space. We also see - green bar - that the average gradient vector computed using a random subset of examples resembles the exact empirical gradient computed using all the training data. This suggests that a random subset provides a reasonable estimate of the true empirical gradient. The picture changes completely when we compute - red bars - the distance between the average gradient corresponding to one of the 3 transfer *scoring functions*, and the average random gradient or the empirical gradient. Now the distances are rather large, which suggests that CL by transfer indeed stirs the weights towards different local minima in parameter space as compared to *vanilla* training.

---

[8]In this experiment $g_{F_\theta}(1)$ was set such that it corresponds to $10\%$ of the data or 250 examples. This number was set arbitrarily, with similar qualitative results obtained for a large range of other choices.

[9]As customary, total variance denotes the trace of the covariance matrix.

We see in Fig. 6b that the total variance for the 3 transfer *scoring functions* is much smaller than the total variance of some random subset of the whole training set. This intuitive result demonstrates the difference between training with easier examples and training with random examples, and may – at least partially – explain the need for a different learning rate when training with easier examples.

## 3.4 ALTERNATIVE PACING FUNCTIONS

**Single step pacing.** Curriculum learning can be costly, and it affects the entire learning protocol via the *pacing function*. At the same time, we note that the main effect of the procedure takes place at the beginning of training. This empirical observation may be due, in part, to the fact that the proposed *scoring function f* is based on transfer from another network trained on a different dataset, which only approximates the unknown *ideal scoring function*. Possibly, since the *scoring function* is based on one local minimum in a complex optimization landscape which contains many local minima, the score given by $f$ is more reliable for low scoring (easy) examples than high scoring (difficult) examples, that may be in the vicinity of a different local minimum.

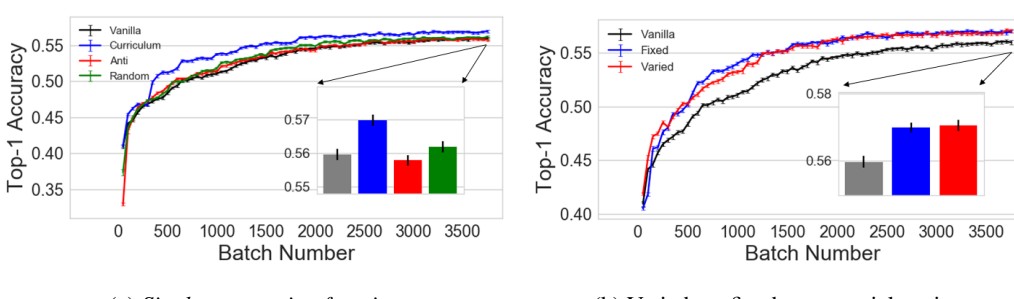

(a) *Single step pacing function*                    (b) Varied vs. fixed exponential pacing.

Figure 7: Results in **case 1**, with Inception-based *transfer scoring function*. (a) *Single step pacing function*, (b) *varied Exponential pacing function*. Inset: bars indicating the average final accuracy in each condition over the last few iterations. Error bars indicate the STE after 50 repetitions.

Once again we evaluate **case 1**, using the *transfer scoring function* and the *single step pacing function*. We see improvement in the test accuracy in the curriculum test condition which resembles the improvement achieved using the *exponential pacing*. Results are shown in Fig. 7a. It is important to note that this *pacing function* ignores most of the prior knowledge provided by the *scoring function*, as it only uses a small percent of the easiest examples, and yet it achieves competitive results. Thus we see that in our empirical setup, most of the power of CL lies at the beginning of training.

**Varied exponential pacing.** This *pacing function* allows us to run a CL procedure without the need for further tuning of learning rate. Once again we evaluate **case 1**, fixing the learning rate parameters to be the same as in the *vanilla* test condition, while tuning the remaining hyper-parameters as described in Section 2.3 using a grid search with cross-validation. We see improvement in the accuracy throughout the entire learning session, although smaller than the one observed with *fixed exponential pacing*. However, decreasing the learning rate of the *vanilla* by a small fraction and then tuning the curriculum parameters achieves results which are very similar to the *fixed exponential pacing*, suggesting that this method can almost completely nullify the indirect manipulation of the learning rate in the *fixed exponential pacing* function. These results are shown in Fig. 7b.

## 3.5 SUMMARY OF RESULTS

Fig. 8 summarizes the main results presented in the paper, including: curriculum with an Inception-based *scoring function* for (i) *fixed* exponential pacing (denoted *curriculum*), (ii) *varied* exponential pacing, and (iii) *single step* pacing. It also shows curriculum with fixed exponential pacing for (iv) *self-paced* scoring, and (v) *self-taught* scoring. In addition, we plot the control conditions of *vanilla*, *anti*-curriculum, and *random*. In Fig. 8a we see the learning curves of the above conditions, with inset bars that depict the final accuracy of each condition, and error bars that represent the standard error after 50 repetitions. All the curriculum conditions seem to improve the learning accuracy throughout the entire learning session while converging to similar performance, excluding the self-paced *scoring function* which impairs learning.

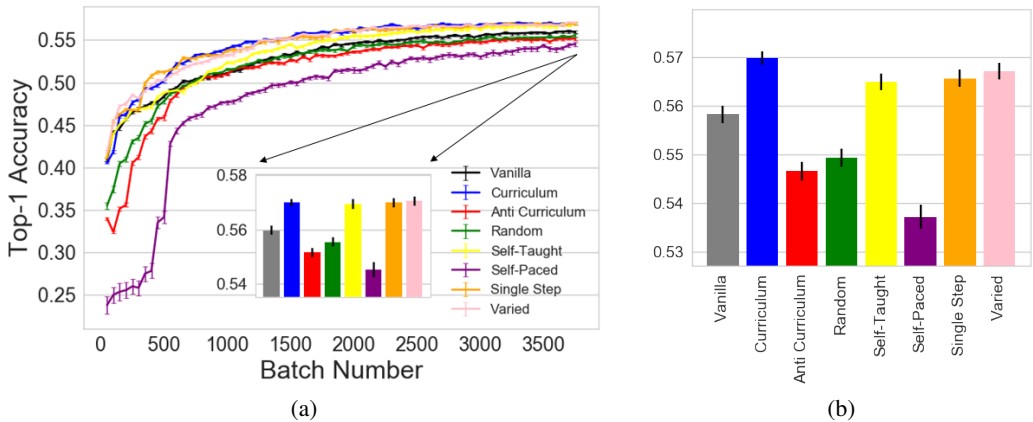

Figure 8: Results in **case 1**, showing the learning curves for all different test conditions. (a) Inset: bars indicating the average final accuracy in each condition over the last few iterations. (b) Bars indicate the final accuracy when picking the parameters which maximize the area under the curve.

The learning curves shown in Fig. 8a were obtained by searching for the parameters that maximize the final accuracy. This procedure only takes into account a few data points, which makes it less robust. In Fig. 8b we plot the bars of the final accuracy of the learning curves obtained by searching for the parameters that maximize the Area Under the Learning Curve. AUC is positively correlated with high final performance while being more robust. Comparing the different conditions using this maximization criterion gives similar qualitative results - the performance in all the curriculum conditions is still significantly higher than the control conditions. However, now the curriculum based on the Inception-based *scoring function* with *fixed exponential pacing* achieves performance that is significantly higher than the other curriculum methods, in evidence that it is more robust.

## 4 SUMMARY AND DISCUSSION

Above we formally defined a curriculum learning algorithm, decomposing it into two separate problems: (i) How to determine the difficulty of the training data (via the *scoring function*)? (ii) At which pace should the learner be shown the more advanced data (via the *pacing function*)? We defined a *scoring function* based on transfer learning from a large network, showing that it can both speed up the rate of learning and improve the final accuracy. This was shown using a number of test cases, including particularly challenging subsets of CIFAR 100 and ImageNet datasets, and the entire CIFAR-10 and CIFAR-100 datasets. We used both a relatively small hand-crafted CNN, and a large public-domain completive VGG-based network. We observed that most of the beneficial effect of CL was achieved at the beginning of the learning and that the benefits were more significant when using harder datasets. During our experiments, we saw that the quality of the teacher network also impacts curriculum learning by transfer. In order for the teacher network to differentiate between easier and harder examples, it should have reasonable generalization accuracy. A teacher with low performance will classify all points as hard, while a "too good" teacher will classify all points as easy, results in a less efficient curriculum. Based on these observations, we investigated two alternative *pacing functions* that achieved CL with less overhead as compared to training without a curriculum.

In addition to the *transfer scoring function*, we introduced the *self-taught scoring function*. This function does not rely on transfer from a large network, and can therefore, presumably, better scale up to larger datasets. *Self-Taught scoring* is closely related to Self Paced Learning, yet it boils down to essentially the opposite scoring heuristics, since the *self-taught scoring function* relies on the final hypothesis of a pre-trained network while SPL relies on the current hypothesis. In agreement with the theory reviewed in the introduction, we showed that the *self-paced scoring function* impaired the learning, while the *self-taught scoring function* enhanced it. In other words, when choosing easier points to guide the learning, it is important to measure difficulty with respect to the final hypothesis, not the current hypothesis.

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

## APPENDIX

## A ADDITIONAL EMPIRICAL RESULTS

**CL with other CIFAR-100 super-classes.** In Section 3 we present results when learning to discriminate the "small mammals" superclass of CIFAR-100. Similar results can be obtained for other super-classes of CIFAR-100, including the super-classes of "people", "insects" and "aquatic mammals". CL trained on these different super-classes shows the same qualitative results. We note once again that CL is more effective in the harder tasks, namely, the super-classes containing classes that are harder to discriminate (as seen by lower *vanilla* accuracy). As an example, Fig. 9 shows results using the "aquatic mammals" dataset, which greatly resembles the results we've seen when discriminating the "small mammals" dataset (cf. Fig.8).

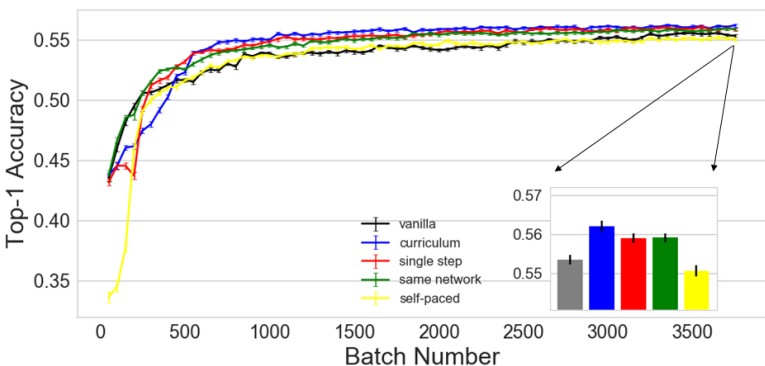

Figure 9: Results under the same conditions as in Fig. 8b, using instead the "aquatic mammals" CIFAR-100 superclass. Error bars show STE after 50 iterations.

**Transfer based scoring function.** In the experiments described in Section 3, when using the *transfer scoring function* defined in Section 2.3, we use the pre-trained Inception network available from https://github.com/Hvass-Labs/TensorFlow-Tutorials. We first normalized each training image to the range $[-1, 1]$, resized it, and ran it through the Inception network. We then used the penultimate layer's activations as features for each training image, resulting in 2048 features per

image. Using these features, we trained a Radial Basis Kernel (RBF) SVM (Scholkopf et al., 1997) and used its confidence score to determine the difficulty of each image. The confidence score of the SVM was provided by *sklearn.svm.libsvm.predict_proba* from Python's Sklearn library and is based on cross-validation.

Choosing Inception as the teacher and RBF SVM as the classifier was a reasonable arbitrary choice – the same qualitative results are obtained when using other large networks trained on ImageNet as teachers, and other classifiers to establish a confidence score. Specifically, we repeated the experiments with a *transfer scoring function* based on the pre-trained VGG-16 and Resnet networks, which are also trained on Imagenet. The curriculum method using the *transfer scoring function* and *fixed exponential pacing function* are shown in Fig. 10a, demonstrating the same qualitative results. Similarly, we used a linear SVM instead of the RBF kernel SVM with similar results, as shown in Fig. 10b. We note that the STE error bars are relatively large for the control conditions described above because we only repeated these conditions 5 times each, instead of 50 in the main experiments.

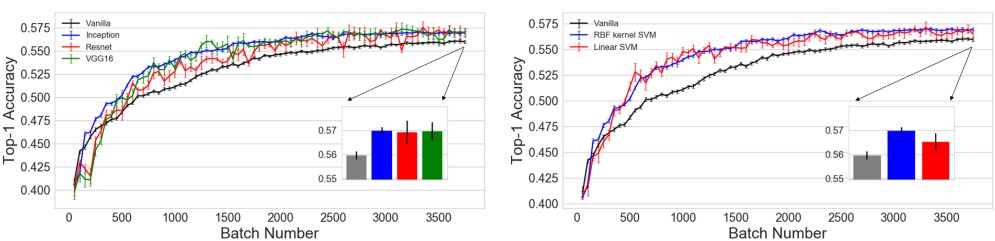

(a) Three competitive networks trained on Imagenet.    (b) Two different classifiers.

Figure 10: Results in **case 1**, comparing different variants of the *transfer scoring function*. The inset bars show the final accuracy of the learning curves. The error bars shows STE after 50 repetitions for the *vanilla* and Inception conditions with *RBF kernel SVM*, and 5 repetitions for the *Resnet*, *VGG-16* and the *Linear SVM* conditions.

## B  EXTENDED DISCUSSION

**Self-taught bootstrapping**    In principle, the *self-taught scoring function* can be used repeatedly to boost the performance of the network indefinitely: after training the network using a curriculum, we can use its confidence score to define a new *scoring function* and retrain the network from scratch. However, *scoring functions* created by repeating this procedure tend to accumulate errors: once an example is misclassified as being easy, this example will be shown more often in subsequent iterations, making it more likely to be considered easy. In practice, we did not observe any benefit to repeated bootstrapping, and even observed the impairment after a large number of repetitions.

### FAIR COMPARISON IN PARAMETER TUNING

When using the moderate size hand-crafted network (**cases 1, 2** and **3**), learning rate tuning is done for the *vanilla* case as well. In these cases, for the *curriculum*, *anti-curriculum* and *random* test conditions, we perform a coarse grid search for the *pacing* hyper-parameters as well as the learning rate hyper-parameters, with an identical range of values for all conditions. For the *vanilla* condition, there are no *pacing* hyper-parameters. Therefore, we expand and refine the range of learning rate hyper-parameters in the grid search, such that the total number of parameter combinations for each condition is approximately the same.

When using a public domain competitive network (**case 4**), the published learning rate scheduling is used. Therefore we employ the *varied exponential pacing function* without additional learning rate tuning and perform a coarse grid search on the *pacing* hyper-parameters. To ensure a fair comparison, we repeat the experiment with the *vanilla* condition the same number of times as in the total number of experiments done during grid search, choosing the best results. The exact range of values that are used for each parameter is given below in Appendix C. All prototypical results were

confirmed with cross-validation, showing similar qualitative behavior as when using the coarse grid search.

To control for the possibility that the results we report are an artifact of the way the learning rate is being scheduled, which is indeed the method in common use, we test other learning rate scheduling methods, and specifically the method proposed by Smith (2017) which dynamically changes the learning rate, increasing and decreasing it periodically in a cyclic manner. We have implemented and tested this method using **cases 2** and **3**. The final results of both the *vanilla* and *curriculum* conditions have improved, suggesting that this method is superior to the naïve exponential decrease with grid search. Still, the main qualitative advantage of the CL algorithm holds now as well - CL improves the training accuracy during all stages of learning. As before, the improvement is more significant when the training dataset is harder. Results for **case 3** (CIFAR-100) are shown in Fig. 11.

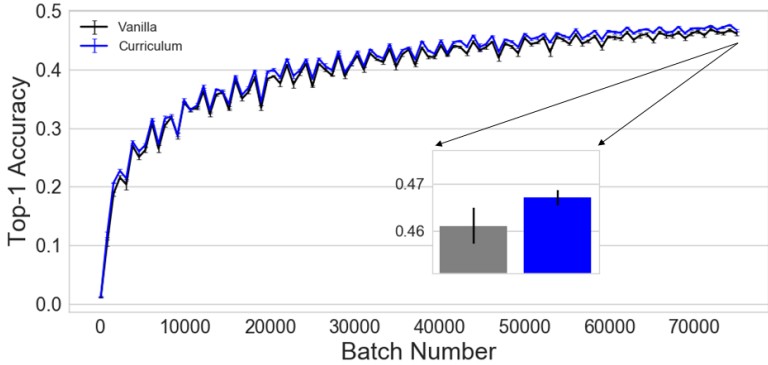

Figure 11: Results under conditions similar to empirical **case 3** as shown in 3b, using cyclic scheduling for the learning rate as proposed by Smith (2017).

## C  METHODOLOGY, ADDITIONAL DETAILS

**Exponential Pacing**   Throughout this work, we use *pacing functions* that increase the data size each *step* exponentially. This is done in line with the customary change of learning rate in an exponential manner.

**Architecture Details**   The moderate-size neural network we used for **cases 1,2,3**, is a convolutional neural network, containing 8 convolutional layers with 32, 32, 64, 64, 128, 128, 256, 256 filters respectively. The first 6 layers have filters of size $3 \times 3$, and the last 2 layers have filters of size $2 \times 2$. Every second layer there is a $2 \times 2$ max-pooling layer and a $0.25$ dropout layer. After the convolutional layers, the units are flattened, and there is a fully-connected layer with 512 units followed by $0.5$ dropout layer. Batch size was 100. The output layer is a fully connected layer with 5 output units, followed by a softmax layer. We trained the network using the SGD optimizer, with cross-entropy loss. All the code will be published upon acceptance.

**Grid-search hyper parameters**   When using grid search, identical ranges of values are used for the *curriculum, anti-curriculum* and *random* test conditions. Since *vanilla* contains fewer parameters to tune – as it has no *pacing* parameters – we used a finer and broader search range. The range of parameters was similar between different *scoring functions* and *pacing functions* and was determined by the architecture and dataset. The range of parameters for **case 1**: (i) initial learning rate: $0.1 \sim 0.01$; (ii) learning rate exponential decrease $2 \sim 1.1$; (iii) learning rate *step size* $200 \sim 800$; (iv) *step size* $20 \sim 400$, for both varied and fixed; (v) *increase* $1.1 \sim 3$; (vi) *starting percent* $4\% \sim 15\%$ (note that $4\%$ is in the size of a single mini-batch). For **cases 2, 3** the ranges is wider since the dataset is larger: (i) initial learning rate: $0.2 \sim 0.05$; (ii) learning rate exponential decrease $2 \sim 1.1$; (iii) learning rate *step size* $200 \sim 800$; (iv) *step size* $100 \sim 2000$, for both varied

and fixed; (v) *increase* $1.1 \sim 3$; (vi) *starting percent* $0.4\% \sim 15\%$. For **cases 4**, the learning rate parameters are left as publicly determined, while the initial learning rate has been decreased by $10\%$ from $0.1$ to $0.09$. The *pacing* parameter ranges are: (i) *step size* $50 \sim 2500$, for both varied and fixed; (ii) *increase* $1.1 \sim 2$; (iii) *starting percent* $2\% \sim 20\%$.

**ImageNet Dataset Details**    In **case 5**, we used a subset of the ImageNet dataset ILSVRC 2012. We used 7 classes of cats, which obtained by picking all the hyponyms of the cat synset that appeared in the dataset. The 7 cat classes were: 'Egyptian cat', 'Persian cat', 'cougar, puma, catamount, mountain lion, painter, panther, Felis concolor', 'tiger cat', 'Siamese cat, Siamese', 'tabby, tabby cat', 'lynx, catamount'. All images were resized to size $56 \times 56$ for faster performance. All classes contained 1300 train images and 50 test images.

