# OpenReview forum: "In Your Pace: Learning the Right Example at the Right Time"
_ICLR.cc/2019/Conference_

### Official Review · AnonReviewer1 · 2018-10-29
**Investigates an interesting problem but has limited novelty and presents limited insights**

**Rating:** 4
**Confidence:** 4

**Review:**

This problem of interest in this paper is Curriculum Learning (CL), in the context of deep learning in particular. CL refers to learning a non-random order of presenting the training examples to the learner, typically with easier examples presented before difficult ones, to guide learning more effectively. This has been shown to both speed up learning and lead to better generalization, especially for more challenging problems. In this paper, they claim that their contribution is to decompose the problem of CL into learning two functions: the scoring function and the pacing function, with the role of the former being to estimate the difficulty of each training example and the latter to moderate the schedule of presenting increasingly more challenging examples throughout training.

Overall, I found it hard to understand from reading the paper what exactly is new versus what is borrowed from previous work. In particular, after reading Weinshall et al, I realized that they have already proposed a number of things that are experimented with here: 1) they proposed the approach of transfer learning from a previously-trained network as a means of estimating the ‘scoring function’. 2) they also distinguish between learning to estimate the difficulty of examples, and learning the schedule of decreasing difficulty throughout learning, which is actually stated here as the contribution of this paper. In particular, in Section 3 of Weinshall et al, there is a sub-section named “scheduling the appearance of training examples” where they describe what in the terminology of this paper would be called their pacing function. They experiment with two variants: fixed, and adaptive, which are very similar to two of the pacing functions proposed here.

Bootstrapping:
A component of this work that didn’t appear in Weinshall et al, is the bootstrapping approach to estimating the scoring function. In general, this involves using the same network that is being trained on the task to estimate the difficulty of the training examples. The authors explain that there are two ways to do this: estimate how easy each training example is with respect to the ‘current hypothesis’ (the weights of the network at the current step), and with respect to the ‘final hypothesis’, which they estimate if I understand correctly as the network at the end of training. The latter would necessitate first training the network in the standard way, and then using it to estimate how easy or hard each example is, and using those estimates to re-train the network from scratch using that curriculum. They refer to the former as self-paced learning and to the latter as self-taught learning. I find these names confusing in that they don’t really convey what the difference is between the two. Further, while self-paced learning has been studied before (e.g. Kuman et al), I’m not sure about self-taught learning. Is this a term that the authors here coined? If not, it would be useful to add a reference.

Using easy / hard examples as judged by the current / final hypothesis:
When using the current hypothesis, under some conditions, Weinshall et al showed that choosing harder examples is actually more beneficial than easy examples, similar in spirit to hard negative mining. On the other hand, when using the final hypothesis to estimate examples’ difficulty, using a schedule of increasing difficulty is beneficial. Based on this, I have two comments: 1) It would therefore be useful to implement a version that uses the current hypothesis to estimate how easy each example is (like the self-paced scoring function) but then invert these estimates, in effect choosing the most challenging instead of the easiest ones as is done for anti-curriculum learning. This would be a hybrid between the current self-paced scoring function and anti-curriculum scoring function that would essentially implement the hard negative mining technique in this context. 2) It would be useful to comment on the differences between the self-paced scoring function used here, and that in Kumar et al. In particular, in this case using a curriculum based on this scoring function seems to harm training but in Kumar et al, they showed it actually increased performance in a number of different cases. Why does one work but the other doesn’t?

Experiments:
The experiments are presented in a subset of 5 classes from CIFAR-10 (also used by Weinshall et al.), but also in the full CIFAR-10 and CIFAR-100 datasets. They used both a small CNN (same as in Weinshall et al) as well as a VGG architecture. Overall, their results are comparable to what was previously known: using a curriculum computed by transfer leads to improved learning speed and final performance (though sometimes very slightly) compared to the standard training, and the training with a random curriculum. Further, the benefit is larger when the task is harder (as measured by the final vanilla-trained performance). By computing the distances between the gradients obtained from using a curriculum (via the transfer scoring function) and no curriculum confirms that these two training setups indeed drive the learning in different directions; an analysis similar to Weinshall et al. Also, since, as was previously known and they also observe, the benefit of CL is larger at the beginning of training, they propose a single-step pacing function that performs similarly to other pacing functions while is simpler and more computationally effective. The idea is to decrease only once the proportion of easy examples used in mini-batches, via a step function. Therefore at the start many easy examples are used, and after this threshold is surpassed, few easy examples are used.

Overall, I don’t feel the contribution of this paper is large enough to recommend acceptance. The main points that guided this decision are:
1) The relationship with previous work is not clear. In particular, Weinshall et al seems to have already proposed a few components that are claimed to be the contribution of this paper, as elaborated on above. The authors should mention that the transfer scoring function was borrowed from Weinshall et al, clarify the differences between their pacing functions from those in Weinshall et al., etc.
2) The usefulness of using easy or hard experiments when consulting the current or final hypothesis is discussed but not explored sufficiently. An additional experiment is proposed above to add another ‘data point’ to this discussion.
3) self-paced learning is presented as something that doesn’t work and wasn’t expected to work. However, in the past successes were shown with this method, so it would be useful to clarify the difference in setup, and justify this difference.
4) It seems that the experiments resulted to similar conclusions to what was already known. While it’s useful to confirm these findings on additional datasets, I didn’t feel that there was a significant insight gained from them.

---

### Official Review · AnonReviewer2 · 2018-11-01
**lacking convincing comparison**

**Rating:** 4
**Confidence:** 4

**Review:**

This paper studies an interesting and meaningful topic that what is the potential of curriculum learning (CL) in training dnn.  The authors decompose CL into two main parts: scoring function and pacing function. Towards both parts, several candidate functions are proposed and verified.  The paper is presented quite clear and gives contribution to better understand CL in the literature of DNN.

However, I have several concerns towards the status of this paper.

First, quite a few important related works are missing by the authors. Just name a few, [1] studies designing data curriculum by predictive uncertainty. [2,3] studies how to derive data driven curriculum along NN training. In particular, the objective of [2] is exactly “learning the right examples at the right time”. All these three papers focus on, or at least talk about, neural network training. Unfortunately, none of them are compared with, or even referenced.

Second, although comprehensive study towards different curriculum strategy are given, I found it largely unconvincing. I tried hard to discover a *detailed accuracy number on a benchmark dataset with unchanged setting* but found only case 4. By ‘unchanged’ I mean it is not a subpart of the whole dataset, or using a rarely seen nn architecture.  If it is such `changed’ settings, the results are largely unconvincing since we do not know what the exact baseline is. For the only ‘unchanged’ setting 4 including VGG on CIFAR100, unfortunately the results seem not good (Fig 4a). I understand that some previous work such as the cited [Weinshall et.all 2018] also used the same setting: however it does not mean such settings give *clear and convincing* results of whether CL plays significant role in training DNN. Furthermore, I also expect the results of comparing in terms of wall clock time (including all your bootstrapping training time) but not merely batch numbers.

[1] Chang, Haw-Shiuan, Erik Learned-Miller, and Andrew McCallum. "Active Bias: Training More Accurate Neural Networks by Emphasizing High Variance Samples." NIPS. 2017.

[2]  Fan, Y., Tian, F., Qin, T., Li, X. Y., & Liu, T. Y. Learning to Teach. ICLR 2018

[3] Jiang, Lu, et al. "MentorNet: Learning data-driven curriculum for very deep neural networks on corrupted labels." ICML. 2018.

---

### Official Review · AnonReviewer3 · 2018-11-02
**a good start**

**Rating:** 5
**Confidence:** 4

**Review:**

In my opinion this paper is generally of good quality and clarity, modest originality and significance.

Strengths:
- The experiments are very thorough. Hyperparameters were honestly optimized. The method does show some modest improvements in the experiments provided by the authors.
- The analysis of the results is quite insightful.

Weaknesses:
- The experiments are done on CIFAR-10, CIFAR-100 and subsets of CIFAR-100. These were good data sets a few years ago and still are good data sets to test the code and sanity of the idea, but concluding anything strong based on the results obtained with them is not a good idea.
- The authors claim the formalization of the problem to be one of their contributions. It is difficult for me to accept it. The formalization that the authors proposed is basically the definition of curriculum learning. There is no novelty about this.
- The proposed method introduces a lot of complexity for very small gains. While these results are scientifically interesting, I don't expect it to be of practical use.
- The results in Figure 3 are very far from the state of the art. I realize that they were obtained with a simple network, however, showing improvements in this regime is not that convincing.  Even the results with the VGG network are very far from the best available models.
- I suggest checking the papers citing Bengio et al. (2009) to find lots of closely related papers.

In summary, it is not a bad paper, but the experimental results are not sufficient to conclude that much. Experiments with ImageNet or some other large data set would be advisable to increase significance of this work.

---

### Public Comment · (anonymous) · 2018-10-08
**It is a meaningful work, but there are a few questions**

1. The results of cifar10 and cifar100 are very low as shown in Case 2 and Case 3. Do you conduct a data augmentation in your experiments? If you did, is the score of one image changing?
2. The setting of learning rate should provide more detailed explanation. How long is the cyclic period? If you can show the learning rare and your pace function in one fig, this will be great.

---

> ### Author Response · Authors · 2018-10-10
> **Answers**
>
> Thank you for your comment.
>
> 1. Case 2 and Case 3 are performed on the same moderate size network described in Case 1, which was not constructed with cifar10 and cifar100 in mind - hence the low performance. In case 4, we used a competitive architecture for CIFAR-100, with a much better performance (results can be seen in Fig. 4.). No data augmentation was used in any of the experiments described in the paper.
>
> 2. In all cases, the learning rate was decreased exponentially every fixed number of iterations (similarly to the way we increase the data size, depicted in Fig. 1.). The use of cyclic learning rate (as in Fig. 10 in the appendix) was done as a control measure only, with a cyclic period calculated as suggested in Smith (2017). We will add to the appendix graphs that show qualitatively these learning rate scheduling functions, for clarification.

---

### Author Response · Authors · 2018-10-21
**Addiotional empirical results on ImageNet**

We've added empirical results on a subset of 7 classes from Imagenet, using the same architecture as cases 1, 2, 3 for both the curriculum by transfer and vanilla test cases.
The results show same qualitative behavior as the previous cases: curriculum by transfer reaches higher accuracy faster and converges to a better final solution.

The paper will be updated once we will be allowed to modify it.

---

### Author Response · Authors · 2018-11-26
**Thank you for your reviews!**

Following the reviews, we've added a section showing that curriculum by transfer achieves similar qualitative improvements to network generalization also when trained on a subset of the popular ImageNet dataset.
We've included a broader review of the relevant literature, emphasizing the difference between previous works and ours.

---

### Meta-Review · Area_Chair1 · 2018-12-13

**Confidence:** 5
**Recommendation:** Reject

**Metareview:**

This paper presents an interesting strategy of curriculum learning for training neural networks, where mini-batches of samples are formed with a gradually increasing level of difficulty.
While reviewers acknowledge the importance of studying the curriculum learning and the potential usefulness of the proposed approach for training neural networks, they raised several important concerns that place this paper bellow the acceptance bar: (1) empirical results are not convincing (R2, R3); comparisons on other datasets (large-scale) and with state-of-the-art methods would substantially strengthen the evaluation (R3); see also R2’s concerns regarding the comprehensive study; (2) important references and baseline methods are missing – see R2’s suggestions how to improve; (3) limited technical novelty -- R1 has provided a very detailed review questioning novelty of the proposed approach w.r.t. Weinshall et al, 2018.
Another suggestions to further strengthen and extend the manuscript is to consider curriculum and anti-curriculum learning for increasing performance (R1).
The authors provided additional experiment on a subset of 7 classes from the  ImageNet dataset, but this does not show the advantage of the proposed model in a large-scale learning setting.
The AC decided that addressing (1)-(3) is indeed important for understanding the contribution in this work, and it is difficult to assess the scope of the contribution without addressing them.